# Evaluation of the Impact of Oral Health on the Daily Activities of Users of the National Health System

**DOI:** 10.3390/ijerph21010092

**Published:** 2024-01-14

**Authors:** Adriane Wood, Antonio Pereira, Enoque Araújo, Júlia Ferigatto, Luisa Buexm, Eliane Barroso, Fabiana Vazquez

**Affiliations:** 1Institute of Education and Research (IEP), Barretos Cancer Hospital, Barretos 14784-400, Brazil; dricwood@gmail.com (A.W.); juliaferigatto25@gmail.com (J.F.); labuexm@gmail.com (L.B.); 2Department of Health Sciences and Children’s Dentistry, Faculty of Dentistry of Piracicaba (FOP), State University of Campinas (UNICAMP), Piracicaba 13414-903, Brazil or apereira@fop.unicamp.br (A.P.); or e265727@dac.unicamp.br (E.A.); 3Department of Dentistry, Faculty of Dentistry, University Center of the Barretos Educational Foundation (UNIFEB), Barretos 14783-226, Brazil; embarroso@uol.com.br; 4Center for Research and Prevention in Molecular Oncology (CPOM), Barretos Cancer Hospital, Barretos 14784-400, Brazil

**Keywords:** oral health, national health strategies, quality of life, unified health system, surveys and questionnaires

## Abstract

Background: the integration of dentistry services in the Unified Health System in Brazil (SUS) is essential in primary care assistance. Objective: we aimed to develop a tool for improving demand flowby evaluating the impact of oral health on the daily activities of users of the Family Health Unitusing the Oral Impacts of Daily Performance (OIDP)tool. Methods: In Barretos, Brazil, a cross-sectional study was conducted at a Family Health Unit (FHU)including patients over 12 years old. Oral health impact was assessed using the Oral Impacts of Daily Performance (OIDP) tool, and family risk was measured with the Coelho–Savassi scale. Results: 430 participants, including 411 adults and 19 young people, were recruited. Of the adults, 31% had an average OIDP score of 16.61. For young people, 53% reported an impact (average OIDP score: 28.61). Family risk (R1) was prevalent in 57.9% of young people and 53.3% of adults. Among adults, different activities were affected by risk: smiling without embarrassment (risk level 2), enjoying contact with people (risk level 3), and performing one’s job or social role (risk level 1). Emotional state (R3) had the lowest OIDP score (*p* = 0.029). Conclusion: implementation of the OIDP scale in clinical practice enhances healthcare planning and ensures better-quality and equitable services, thus emphasizing comprehensive oral healthcare within the SUS.

## 1. Introduction

Primary Health Care (PHC) replaced the hospital-centric model and established itself as an organizational strategy to respond to the health needs of the population. In 1990, the Community Health Agent Program (CHAP) was created. After its successful experience in curative and preventive care, this program (CHAP) evolved into the Unified Health System (SUS). This program revolutionized the Family Health Unit (FHU), and its guidelines began to oppose those of the previous model of care based on curative and medicalization logic. This FHU program proposed a form of care centered on the family and the territory, consisting of actions for disease prevention, promotion, and health care. Implementation of the FHU has required professionals specialized in primary care, such as home care [1], with skills, abilities, and specialties that differ from those required for focal therapies in a hospital environment.

The home visitation (HV) program began to gain significance in the twentieth century, with sanitary practices brought over from Europe, as a means of promotion and a strategy to combat communicable diseases at that time [2]. The SUS provides home health care to assist those who need continuous care;however, it is mainly used as a tool for local diagnosis and the planning of actions based on individual circumstances [3]. Thus, it emphasizes the importance of home visitation for establishing a relationship with the population. Moreover, its strategic characteristics of integrality and humanization of these actionsallows greater proximity to the population and accountability of professionals for their health needs and social and family life [4,5,6]. In Brazil, after implementation of the National Policies for Basic Care (NPBC), Ordinance No. 2436, of 21 September 2017, a reduction in the number of SUS teams was observed. This reduction led to an increase in social inequality, high demand for care, and shortage of professionals. According to Lucena et al. (2020), in the southern region, the number of teams was reduced by 6.7%, whereas in the northeast region, it was reduced by 4.8% [7].

In this sense, the construction of a public oral health network has been able to minimize the number of oral cancer cases [8] because the provision of dental care via the SUS has proved to be beneficial. Therefore, the need for long-term care and continuity in dental health centersas well as administration fundingneeds to be prioritized [9]. However, despite technological advances and scientific research in the field of dentistry, oral health still represents a major problem in public health [10,11]. A global analysis showed that oral diseases were considered endemic, representing a total of 3.9 billion people worldwide, among whom caries accounts for 35% of cases, despite it being an avoidable disease in comparison with other dental pathologies. For example, severe periodontitis occupies 6th place in the ranking of oral diseases, followed by oral cancer in 11th place and tooth loss in 36th place [12,13].

Thus, the integration of dentistry services in the SUS has become essential in primary care assistance. According to the Ministry of Health (MH), the performance of oral health professionals can be divided into two modalities: Mode I, represented by a dentist and an assistant; and Mode II, with a dentist, an assistant, and a dental hygiene technician [14,15,16]. The actions within the SUS are expressed by criteria such as operational characterization and the definition of the family as the recipient of comprehensive and humanized care [17,18]. Therefore, the incorporation of dentistry into the SUS has positively influenced the population’s access to oral health care. Consequently, risk stratification tools such as the Coelho–Savassi scale(the Family Risk Scale developed by Coelho–Savassi, ERF-CS), whichclassifies the vulnerability, risk factors, and aggravating factors of patients enrolled in the system [19,20,21,22], and tools that determine the impact of oral changes on the daily life of individuals [23], such as the OIDP (Oral Impact of Daily Performance), can help to improve the quality of life of the population [24,25,26,27,28] and are useful for planning health services [29,30,31,32,33].

The aim of this study was to develop a tool for improving demand flow by evaluating the impact of oral health on the daily activities of users of a Family Health Unit using the Oral Impacts of Daily Performance (OIDP).

The demand flow in Primary Health Care (PHC) is characterized by the user’s attitude towards seeking health services to access and resolve their needs. The demand for health services is considered spontaneous when the user arrives at the health unit without having made an appointment; programmed when health services are scheduled; and repressed when people who need care are unable to access the health service.

## 2. Materials and Methods

### 2.1. Study Definition and Inclusion

This was a cross-sectional, prospective, analytical study that was conducted at the Family Health Unit (FHU) of the Dr. Wilson Hayek Saihg facility, located in the Nogueira District of the city of Barretos, São Paulo, Brazil. This Family Health Unit (FHU), designed for 4000 registered patients, is the first point of access to the SUS. Registered users who met the inclusion criteria for eligibility and were over the age of 12 were considered. At the time of inclusion, the participants were informed, orally and in writing, about the research. Exclusion criteria were individuals who did not agree to participate in this study, those with disorders that prevented them from answering questions by themselves, and those who refused to sign the free and informed terms of consent. Participation in the research was conditional on the prior reading and signing of the afore-mentioned free and informed terms of consent form (FITC), respecting the participant’s right to refuse or withdraw at any time during the research, according to the ethical precepts of research involving human beings. It is also noteworthy that the confidentiality of the participants involved was safeguarded.

### 2.2. Sample Calculation and Data Collection

In this study, a stratified sample with optimal allocation was used, and secondary data obtained from the National Oral Health Survey (SB Brazil) were considered. The OIDP was used, which showed 27.9% of participants had their daily activities impacted by oral health problems [34]. The sample calculation performed considered a significance level of 0.05, a power of 0.80, and the 28% of patients experiencing an impact on daily activities arising from oral health problems. Extracts were obtained from 365 R1 families, 32 R2 families, and 33 R3 families, totaling 430 families. The families were randomized via REDCap and invited to participate in this study. After accepting and signing the two-way terms of consent for family members over the age of 18 and terms of consent for those under 18 and over 12, participants were asked to complete the OIDP questionnaire.

The data collection procedure initially proposed for this studyinvolvedobtaining data during home visits. With the onset of the COVID-19 pandemic, recruiting was performed on demand and an active search for participants was conducted at the SUS.

A sociodemographic questionnaire was used to collect patient identification data such as name, date of birth, gender, ethnicity, age, marital status, education, and family income; the ERF-CS was a useful instrument for classifying the vulnerability, and consequently, the risk factors called sentinels, stratified into 3 scores, and aggravating factors of the families registered. The score for each sentinel was proposed by the authors in accordance with the criteria having the greatest impact on health and social life. Thus, 6 sentinels had a score of 3 (bedridden; physical disability; mental disability; low sanitation conditions; severe malnutrition; a resident/room ratio greater than 1); 3 sentinels had a score of 2 (drug addiction; unemployment; a resident/room ratio equal to 1); and 6 sentinels had a score of 1 (illiteracy; individual under 6 months of age; individual over 70 years of age; arterial hypertension; systemic; diabetes mellitus; a resident/room ratio less than 1). The total sum was classified as R1 (5 or 6); R2 (7 or 8); orR3 (greater than 9) [22].The OIDP instrument had a test–retest reliability of 0.69 and a Cronbach’s alpha ranging from 0.69 to 0.6713 and assessed whether participants had had any oral health problems “in the past six months” which had caused difficulties or impairments in the different domains, viz., eating and enjoying food, speaking and pronouncing clearly, and teeth hygiene; psychological: sleeping and relaxing, smiling, laughing, and showing teeth without being embarrassed, and maintaining a balanced emotional state; and social: performing main job or social roleand enjoying being in contact with people [12,24,35].

### 2.3. Data Analysis

The data were collected and stored on the REDCap platform. Subsequently, data were analyzed using measures of central tendency and dispersion for quantitative variables and proportions for qualitative variables. All analyses were performed using the SPSS software version 27.

#### 2.3.1. Part 1—Pre-Production

In the pre-production phase, this study sought to develop a toolbased on the scoring and classification of two other instruments, which were used to assess the risk classification of the population (ERF-CS) and to assess the impact of oral health on daily activities based on the patient’s own self-perception (OIDP). At first, data for the project were collected by screening the participants; therefore, a data sample was collected for the population (based on a pre-established sample calculation for this study). However, due to the onset of the COVID-19 pandemic, we began to collect data on demand;i.e., participants were included by convenience.

The data collection activity was performed before constructing the tool developed in this study, because statistical calculations were made based on the results of the study population for the purpose of constructing the tool. This study was divided into two primary stages: score evaluation and measurement of results with the construction of the tool. In the first, sociodemographic and clinical data were used for the assessment of family risk classification using the ERF-CS and self-perceived assessment of the impact of oral health on daily activities according to the OIDP questionnaire. In the second step, all data collected in step 1 were statistically analyzed.

#### 2.3.2. Part 2—Production

The pre-structured tool construction was based on the assessment of the sum of ERF-CS scores and the scores of the OIDP questionnaire. This combination was based on a statistical analysis in whichdata were dived into terciles and the classification of each result was pre-determined, and the combination of the two evaluations resulted in the score of the tool developed, named Wood. The tool developed was based on a percentage of the population registered at an FHU. Implementation of the new tool would not cause structural or geographic changes, but it would minimize the repressed demand and waiting lists and would help with better management of health services users. This will would the professionals with relief, and consequently, they would be able to provide care with greater efficiency and quality.

#### 2.3.3. Part 3—Post-Production

The tool for managing and maintaining the flow of demand of users registered with the SUS will not be inserted into the health service for the time being, because firstly, it will be necessary to make a projection of the benefits that its future application would provide based on a literature review and cases conducted for this study. The aim of creating the Wood tool, the product of this study, is to offer professional control of the demand for care, with a view to improving the organization of the SUS by considering the users’ needs. In this sense, when the Wood flow is used, it will be possible to determine the real need for care based on family risk and the self-perception of the individual.

## 3. Results

### 3.1. Data Collection

Data collection took place during twotime intervals: the first was characterized by household data collection, according to randomized screening using REDCap, and stratified according to family risk; the second round of collection began in March 2020 due to the COVID-19 pandemicfor the screening of individuals on demand.

A total of 430 individuals participated in this study, with 411 adults and 19 young people. The mean age for young people was 14.8 (SD = 1.77), with a minimum of 12 years and a maximum of 17 years; for adults, the mean age was 50.3 (SD = 17.65), with a minimum of 18 years and a maximum of 96 years. When analyzing family risk, it was observed that R1 was the most frequent category among young people (57.9%) and adults (53.3%).

Among young people, the predominant gender was male (57.9%), while among adults, females were predominant (62.5%); white ethnicity prevailed in both samples. As regards educational level, among young people, the most prevalent status was illiterate/knows how to read and can write/incomplete elementary school (73.7%); in the adult sample, it was complete high school/incomplete higher education (41.6%). Regarding the family income of the adult sample, 29.2% received one minimum wage,38.3% received two minimum wages, and 32.4% of the sample received three or more (Figure 1).

Considering the clinical data, in 100% of the samples, participants had their own toothbrush. Regarding the presence of bleeding gums, the majority did not have this, with only13 young people (68.4%) and 302 adults (73.5%) reporting this. Dental floss was used by 4 (21.1%) young people and 199 (48.4%) adults; toothpick swere used by 6 (31.6%) young people and 78 (19.0%) adults. The presence of mouth sores was found in 2 (10.5%) young people and 10 (2.4%) adults. Analysis of smoking showed that 3 (15.8%) of the young people were smokers, whose ages ranged from 15 to 17 years; among adults, 88 (21.4%) were smokers. The length of time they smokedand other information may be found in Table 1. Among the sample, 8 young people (42.1%) and 156 (38.0%) adults reported that they had been to the dentist in the last six months, with routine care being the predominant reason for the visit for 14 (73.7%) young people, and 255 (62.0%) adults (Figure 1).

### 3.2. Socioeconomic and Clinical Variables

Table 1 shows the distribution of socioeconomic and clinical variables in the young and adult sample. As regards family income, 60.6% of those earning more than three minimum wages were at risk level 1 (Table 1).

When the association between the distribution of family risk classification and sociodemographic and clinical variables of the participants was analyzed, for the young sample, significance was found only for the gender variable (*p* = 0.045). For adults, the majority of those with over 12 years of schooling were in the R1 group (*p* = 0.004) and so were those who flossed (*p* = 0.004) and non-smokers (*p* ≤ 0.0001). Regarding the reason for the consultation, the majority of patients in R1 reported routine care(*p* = 0.041) (Table 2).

### 3.3. Young People’sOIDP Score

In the OIDP assessment, 10 young people (53%) reported an impact of oral health on their daily activities, with an overall mean of 28.61 (SD = 24.43), with a minimum of 2.79 and a maximum of 68.06. The assessment items sleeping and relaxing and maintaining a balanced emotional state showed the highest number of responses. The mean OIDP scores observed ranged from 0 to 9, as follows: working, 9.0 (SD = 0.00; min. 9.00; max. 9.00); smiling, 9.0 (min. 9.00; max. 9.00); keeping in touch with people, 7.75 (SD = 2.50; min. 4.0; max. 9.0); eating, 7.0 (SD = 4.0; min. 1.0, max. 9.0); maintaining a balanced emotional state, 6.86 (SD = 2.67, min. 4.0; max. 9.0); sleeping and relaxing, 6.43 (SD = 3.36; min. 1.0; max. 9.0); and cleaning teeth, 4.50 (SD = 3.32, min. 1.0; max. 9.0). The mean overall OIDP score of the adult sample was 127 (30.9%), with a mean score of 16.61 (SD = 15.74), a minimum of 1 point, and a maximum of 72 points.

### 3.4. Adult OIDP Questionnaire Stratified by Family Risk

Table 3 presents the mean score for each item of assessment of the OIDP questionnaire stratified by family risk level in the adult sample. It was noted that the act of smiling without being embarrassed showed the highest pre-established score, at risk level 2, followed by eating and enjoying foodat risk level 3. The highest score was seen for enjoying being in contact with people, followed by sleeping and relaxing at risk level 1; performing one’s job or social role, maintaining a balanced emotional state, cleaning teeth, and speaking or pronouncing clearly indicated higher scores. With reference to emotional state, people in the R3 group had a lower OIDP score (*p* = 0.029) (Table 3).

### 3.5. The Wood Classification

For the construction of the Wood classification tool, the Coelho–Savassi scale was used to relate the family risk to the OIDP score of the population that presented an impact (OIDP value > 0). To determine the association of family risk with OIDP results, statistical analysis by tertiles was used;therefore, the OIDP scale was divided into three categories: category 1—0 to 8 points; category 2—9 to 19 points; and category 3—≥20 points.

Therefore, after correlating the two scales (family risk and OIDP), we obtained the following results for the Wood classification: W1 = R1 or R2 + OIDP (0–8 points); W2 = R3 + OIDP (0–8 points), R1 or R2 + OIDP (9–19 points), and R1 + OIDP (above 20 points); W3 = R3 + OIDP (9–19 points) and R2 or R3 + OIDP (above 20 points). Combining family risk with the OIDP score enabled the identification of a total of 37 patients belonging to the W1 group (low urgency of need for care), 55 patients belonging to the W2 group (medium urgency of need for care), and 35 patients belonging to the W3 group (high urgency of need for care) (Table 4).

After the Wood classification of the study population (n = 127), (W1 = 37, W2 = 55, W3 = 35), it was possible to prioritize the scheduling of patients with a high urgency of need for dental treatment (Table 4).

## 4. Discussion

The results show the impact that oral health had on the daily activities of individuals from 12 to 96 years of age belonging to the population of Barretos. Combining the Coelho–Savassi Risk Scale with the OIDP scale revealed important information about the impact sof oral health and the possibility of scheduling patients for urgent care without affecting other dental services. There was a higher prevalence among females, which is also the group that seeks out oral health services the most. The proposed risk stratification for dental care can have significant practical implications for managing demand for oral health services, especially if high-risk patients can be seen more quickly. The introduction of the Wood classification to measure the urgency of need for dental care, based on risk and impact sof oral health, offers a practical and potentially effective methodology for managing the demand for dental services. A specific sample of 137 (32%) individuals presenting impacts was analyzed, totaling 127 (31%) adults and 10 (53%) young people. A study conducted in Africa showed an impact of 8.85%. This population was characterized by a need to eradicate pain by means of tooth extraction, differently from the population of this study, which was characterized by a need for tooth preservation [36]. 

The study by Bulgareli et al. (2018) on the 15–65-years-old age group observed oral health impacts on 27.9% of the population, which accounted for 17,560 individuals, whereas the present study involved 430 participants, and impacts were only observed for 32% [34]. The literature highlights the importance of measuring the impact of oral health in some countries, with variation in samples ranging from 8.85% to 73.6% [34,36,37,38,39,40,41]. This disparity regarding oral health impacts may be clarified by the condition of oral health status due to the cultural and socioeconomic contexts of each population.

As regards gender, females have a higher prevalence of oral health impacts, corroborating the findings of several studies [34,40,42,43,44,45], in which it was observed that women showed more concern about their oral health. Women seek health care services 1.9 times more often than men [45] and are more self-critical of their dental appearance than men [46]. The Caucasian population showed a higher prevalence of oral health impacts in this study, which differs from the results found in the literature showing a higher prevalence among non-whites [44,47]. It is noteworthy that this study was conducted within the SUS, and the literature has shown that the non-white population has greater difficulty in accessing and seeking health care [48].

With regard to smoking, it was found that 21% of the total adult population and 16% of the young population were smokers. The study by Pacheco et al. (2014) showed that in their analyzed sample (445 young university students), 6% were active smokers and 2% were former smokers. In this study, smoking participants were shown to be more likely to consider their oral health to be bad [49]. When the education of the adult participants in this study (n = 411) was analyzed, 42% had a low level of education, and more people with this profile belonged to the R1 group (60%), with statistical significance. The study by Fausto et al. (2020) diverged from this finding by showing that there was an association between schooling, individual risk, and oral health, with a greater predominance of the population with a low level of schooling being among the medium- and high-risk groups, which was not seen in the present research [50].

In this study, with regard to family risk, a higher prevalence of low risk was found in the adult population (male, 56.5%, female, 51.4%). According to Jesus et al. (2020), family risk does not assess only the individual, but the whole family set, and this can be a major influencer in a person’s risk classification. Women seek the services of the health system more frequently than men, and even so, a higher percentage of individuals in the R1 group was found in both genders.

As regards ethnicity, a higher percentage of non-white individuals in the R1 group (non-white 54%; white 51.8%) was found. Moreover, a higher frequency of high family risk (R3) was noted among white (29.4%), black (28.0%), and brown (24.1%) individuals [51]. Among the sociodemographic variables related to familial risk, statistical relevance was observed among the variables education, use of toothpicks, use of dental floss, smoking, and reason for consultation. The literature does not show studies that have made the same comparison as this study; however, in isolation, there are studies that have reported these variables as having an influence on the oral health of the individual [34,47,51,52].

With regard to the reason for consultation, the present study classified the reasons asroutine care, emergency, and other. Among the participants who showed oral health impacts, routine care was found to be the primary reason for visits (65.4%) when compared with the other reasons. It could be inferred that people who seek dental care more frequently have a higher self-perception of their oral health, unlike those who delay seeking careor seek it only in cases of emergency [41]. These data were confirmed when the last time the patient was seen by a dentist was analyzed.

The highest OIDP scores were found for the last 12 months only (*p* = 0.009). The study by Vale, Mendes, and Moreira (2013) showed that among the variables of self-perception of the need for treatment, only consultation with a dentist did not prove a statistical association. The other variables influenced the OIDP score, such as frequency of visits to the dentist, type of service provided, reason for the last visit, need for treatment, and the presence of toothache. Furthermore, the highest prevalence of visits to the dentist was for pain, i.e., emergency care, which differs from the findings of the present study [41]. As regards the rating items of the OIDP questionnaire, for those participants who put down a score, there was a predominance in the rating for eating, followed by emotional state and smiling. When correlating this with the different types of family risk, it was observed that only the emotional state classification showed statistical significance (*p* = 0.029).

There are several ways to organize the demand for care, and studies address methodologies that provide effective management of this process, such as risk stratification. This study developed the Wood classification to measure the urgency of dental care, based on risk and the impact of oral health on daily activities. The proposed stratification classified patients into low, medium, and high urgency of need for care, in which the availability of care was high (2 h daily/40 monthly visits), medium (1 h daily/20 monthly visits), and low (1 h daily/20 monthly visits). In this sense, the study by Anghinoni et al. (2021) [52] used the criteria for risk stratification from the Oral Health Guide, measuring and classifying patients as low, medium, and high risk. After the implementation of this stratification, there was an immediate relief regarding the volume of return consultations, because previously, returns were quarterly or at most every six months, and now they are annual for 73% of the population on average. One limitation of this study was that it took place partially during the COVID-19 pandemic, which interfered in data collection, which at first was performed during home visits but was subsequently carried out viacollection on demand.

This study contributed relevant information about the impact of oral health on a specific population and covered a wide age range, thereby providing valuable insights. The combination of the scales offered a thorough understanding of impacts on oral health, and we observed a high prevalence in females, in agreement with previous research. The analysis of smokers identified a relevant risk factor, while the risk stratification proposal has the potential for optimizing dental care. This study had limitations, such as the small sample size of 430 participants, which made it difficult to generalize the results. The COVID-19 pandemic affected data collection, as this was originally performed during home visits and we were forced to undertake on-demand collection instead, which may have impacted the representativeness of the sample, making it subject to bias.

## 5. Conclusions

The conclusion is that creating a demand flow in the management of care at health units is an effective strategy for improving the tracking and monitoring of patients and highlighting the equity, universality, humanization, and quality of the health service offered to the population. This has a positive impact on reducing demand and thus shortening waiting lists and contributes to better planning of prevention, promotion, and health care actions. The analysis showed that the adult population had a higher prevalence of oral health impacts, confirming the importance of a self-perception tool that combines personal assessment with the individual’s socioeconomic assessment. Therefore, the implementation of a demand flow in health care management and the introduction of the Woods cale represent significant advances in the improvement of the health services provided and oral health actions, since they generate a higher number of consultations with better quality and equity of the services provided, promoting the well-being of the community served.

## Figures and Tables

**Figure 1 ijerph-21-00092-f001:**
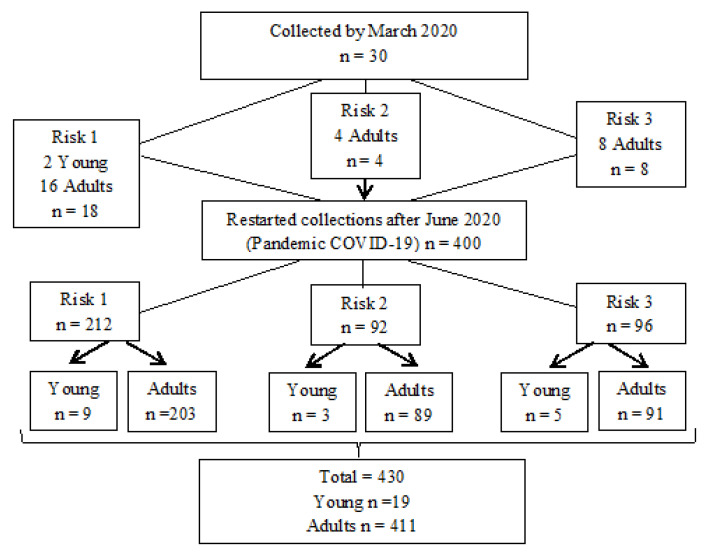
Data collection flow (before and during the COVID-19 pandemic).

**Table 1 ijerph-21-00092-t001:** Socioeconomic and clinical variables of the young (n = 19) and adult (n = 411) sample.

Socioeconomic and Clinical Variables		Young n (%)	Adults n (%)
Age		14.8 (SD = 1.77) ^a^	50.3 (SD = 17.65) ^a^
Family Risk	1	11 (57.9)	219 (53.3)
	2	3 (15.8)	93 (22.6)
	3	5 (26.3)	99 (24.1)
Sex	Female	8 (42.1)	257 (62.5)
	Male	11 (57.9)	154 (37.5)
Ethnicity	White	12 (63.2)	247 (61.9)
	Non-white	7 (36.9)	152 (38.1)
Marital Status	Single	19 (100.0)	107 (26.2)
	Married	-	222 (54.4)
	Widowed	-	43 (10.5)
	Divorced	-	36 (8.8)
Educational Level	<4 years of study	14 (73.7)	140 (34.0)
	<8 years of study	4 (21.1)	65 (15.9)
	<12 years of study	1 (5.3)	171 (41.6)
	≥12 years of study	-	35 (8.5)
Dental Flossing	Yes	4 (21.1)	199 (48.4)
	No	15 (78.9)	212 (51.6)
Tobacco Use	Yes	3 (15.8)	88 (21.4)
	No	16 (84.2)	323 (78.6)
Time of Tobacco Use	Months	1 (33.3)	1 (1.1)
	Years	2 (66.7)	87 (98.9)
Last Visit to the Dentist	Up to 6 months	8 (42.1)	156 (38.0)
	Up to 12 months	5 (26.3)	124 (30.2)
	Up to 24 months	3 (15.8)	38 (9.2)
	Over 24 months	3 (15.8)	93 (22.6)
Reason for Visit	Routine care	14 (73.7)	255 (62.0)
	Emergency	5 (26.3)	154 (37.5)
	Other	-	2 (0.5)
Denture Use	Yes	-	130 (31.6)
	No	-	281 (68.4)
Monthly Family Income	1 minimum wage	-	119 (29.2)
	2 minimum wages ^b^	-	156 (38.3)
	More than 3 minimum wages ^b^	-	132 (32.4)

^a^ Mean. SD = Standard Deviation. ^b^ Minimum wage at the time of data collection = USD255.00.

**Table 2 ijerph-21-00092-t002:** Relationship of familial risk with socioeconomic and clinical variables in the adult sample (n = 411).

Socioeconomic and Clinical Variables		R1n (%)	R2n (%)	R3n (%)	*p*-Value
Sex	Female	132 (51.4)	63 (24.5)	62 (24.1)	0.460
	Male	87 (56.5)	30 (19.5)	37 (24)	
Ethnicity	White	128 (51.8)	57 (23.1)	62 (25.1)	0.574
	Non-white	82 (54)	35 (23)	35 (23)	
Marital Status	Single	47 (43.9)	33 (30.8)	27 (25.2)	0.295
	Married	127 (57.2)	44 (19.8)	51 (23)	
	Widowed	25 (58.1)	8 (18.6)	10 (23.3)	
	Divorced	18 (50)	8 (22.2)	10 (27.8)	
Educational Level	<4 years of study	53 (37.9)	36 (25.7)	51 (36.4)	0.004
	<8 years of study	41 (63.1)	16 (24.6)	8 (12.3)	
	<12 years of study	102 (59.6)	36 (21.1)	33 (19.3)	
	≥12 years of study	23 (65.7)	5 (14.3)	7 (20)	
Dental Flossing	Yes	121 (60.8)	43 (21.6)	35 (17.6)	0.004
	No	98 (46.2)	50 (23.6)	64 (30.2)	
Toothpick Use	Yes	23 (29.5)	25 (32.1)	30 (38.5)	≤0.0001
	No	196 (58.9)	68 (20.4)	69 (20.7)	
Tobacco Use	Yes	34 (38.6)	33 (37.5)	21 (23.9)	≤0.0001
	No	185 (57.3)	60 (18.6)	78 (24.1)	
Last Visit to the Dentist	Up to 6 months	90 (57.7)	32 (20.5)	34 (21.8)	0.335
	Up to 12 months	62 (50)	35 (28.2)	27 (21.8)	
	Up to 24 months	22 (57.9)	5 (13.2)	11 (28.9)	
	Over 24 months	45 (48.4)	21 (22.6)	27 (29)	
Reason for Visit	Routine Care	148 (58.0)	52 (20.4)	55 (21.6)	0.041
	Emergency	71 (46.1)	40 (26)	43 (27.9)	
	Other	0 (-)	1 (50)	1 (50)	
Denture Use	Yes	62 (47.7)	28 (21.5)	40 (30.8)	0.093
	No	157 (55.9)	65 (23.1)	59 (21.0)	
Monthly Family Income	1 minimum wage ^a^	59 (49.6)	31 (26.1)	29 (24.4)	
	2 minimum wages ^a^	80 (51.3)	38 (24.4)	38 (24.4)	0.358
	More than 3 minimum wages ^a^	80 (60.6)	23 (17.4)	29 (22)	

^a^ Minimum wage at the time of data collection = USD255.00.

**Table 3 ijerph-21-00092-t003:** OIDP score.

Variables	Adults							
	R1		R2		R3		*p*-Value	Total
	Mean (SD) ^a^	n (%)	Mean (SD) ^a^	n (%)	Mean (SD) ^a^	n (%)		
A—Eating	5.28 (3.36)	46 (53.4)	6.53 (3.20)	19 (22.1)	5.19 (3.04)	21 (24.5)	0.328	86
B—Talking	5.33 (3.45)	12 (54.5)	3.80 (3.27)	5 (22.7)	3.60 (3.71)	5 (22.7)	0.510	22
C—Cleaning	6.17 (3.05)	18 (56.3)	4.67 (3.61)	6 (18.8)	5.13 (3.44)	8 (25)	0.521	32
D—Sleeping	6.32 (3.04)	19 (48.7)	5.58 (3.20)	12 (30.8)	6.38 (2.33)	8 (20.5)	0.756	39
E—Smiling	7.00 (3.17)	26 (59.1)	8.38 (1.77)	8 (18.2)	7.70 (2.83)	10 (22.7)	0.495	44
F—Emotional state	6.46 (2.87)	28 (50)	6.18 (3.40)	11 (19.6)	3.94 (3.19)	17 (30.4)	0.029	56
G—Performingwork	7.56 (2.66)	16 (48.5)	6.50 (2.67)	8 (24.3)	5.78 (3.93)	9 (27.3)	0.384	33
H—Contact with people	6.52 (3.36)	23 (63.9)	6.50 (2.89)	4 (11.1)	7.89 (2.20)	9 (25)	0.523	36
General	17.24 (17.27)	57 (44.9)	16.70 (13.29)	27 (21.6)	15.19 (14.53)	43 (33.9)	0.881	127

^a^ SD = Standard Deviation.

**Table 4 ijerph-21-00092-t004:** Construction of the Wood classification.

Variable		Overall OIDP for Adult Tertiles			Total
		From 0 to 8	From 9 to 19	Up to 20	
Family Risk	1	^a^ W1 (27)	^b^ W2 (16)	W2 (20)	63
	2	W1 (10)	W2 (6)	^c^ W3 (10)	26
	3	W2 (13)	W3 (12)	W3 (13)	38
Total		58	34	43	127

^a^ W1 = low urgency of need for care. ^b^ W2 = medium urgency of need for care. ^c^ W3 = high urgency of need for care.

## Data Availability

The data presented in this study are available on request from the corresponding author. The data are not publicly available due to the participants’ right to anonymity (confidentiality) as guaranteed in the terms of consent signed by the participants and the researchers.

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
