# Peer review of "Evaluation of the Impact of Oral Health on the Daily Activities of Users of the National Health System"

_ijerph, 2024, doi:10.3390/ijerph21010092_

Round 1

Reviewer 1 Report

Comments and Suggestions for Authors

Line 23 comma instead of full stop

demand flow What do you mean?

430 participants, 411 adults, and 10 young people were recruited. Not clear

Line 40 broke away What do you mean?

Line 96 improve English 

Why analyze few young people and so many adults? 

This is a risk of bias.

Comments on the Quality of English Language

English needs to be improved. Some sentences are difficult to understand.

Author Response

Thank you for your valuable suggestions.

In the text, the requested revisions are in red and the English corrections are in blue

Comments and Suggestions for Authors:

  • Line 23 comma instead of full stop.

Response: We have corrected line 23.

  • “demand flow” What do you mean?

Response: To answer this question, the following was included in the text in lines 24 to 26: “The aim of this study was to develop a tool to improving of demand flow, by evaluating the impact of oral health on the daily activities of users of the Family Health Unit, by using the Oral Impacts of Daily Performances (OIDP).”

To answer this question, the following was included in the text in lines 90 to 97: “The aim of this study was to develop a tool to improving of demand flow, by evaluating the impact of oral health on the daily activities of users of the Family Health Unit, by using the Oral Impacts of Daily Performances (OIDP).

The demand flow in primary health care (PHC) is characterized by the user's attitude towards seeking health services to access and resolve their needs. The demand for health services can be spontaneous, when the user arrives at the health unit without having made an appointment; programmed, when health services are scheduled; and repressed, when people who need care are unable to access the health service.”

  • 430 participants, 411 adults, and 10 young people were recruited. Not clear

Response: It has been corrected: 430 participants were recruited, 411 adults and 19 young people. (line 29)

  • Line 40 broke away What do you mean?

Response: In line 41, we have corrected the word "broke away" to "replaced", so that the text is better understood. (Line 41)

  • Line 96 improve English.

Response: The English of the entire text has been revised as requested.

  • Why analyze so few young people and so many adults? This is a risk of bias.

Response: The fact that there are more adults than young people is due to the fact that when we started the study, our approach was through home visits where all residents would be included in order to classify the risk of families. However, due to the COVID-19 pandemic, in order to maintain the continuity of the study, data collection began at the Family Health Unit. This change explains the lower number of young participants. It is worth noting that we took measures to avoid any bias, although we are aware of this limitation of the study.

  • Comments on the quality of the English language. English needs to be improved. Some sentences are difficult to understand

Response: The English of the entire text has been revised as requested.

*The National Health Strategy (NHS) was corrected in the revision of the English translation to Unified Health System (SUS).

Reviewer 2 Report

Comments and Suggestions for Authors

The manuscript analyzes the impact of oral health on daily activities in a population sample from Brazil. The justification of the study seems to be targeting the benefit that a NHS support could determine in primary dental care especially in deprived areas. The originality of the observation is not supported by a clear methodology. The planned enrolment seems to turn up in contingency plans and the overall data are gathered from stratified groups of the population with an unequal allocation by age. The primary and secondary outcomes are not clear and the discussion doesn't flag properly what the planned actions could be in the light of the observed results.

The paper shows a huge effort from an epidemiological point of view, but the planned actions in terms of public health promotions are not properly presented or explained.

Comments on the Quality of English Language

Major changes

Author Response

Thank you for your valuable suggestions.

In the text, the requested revisions are in red and the English corrections are in blue

Comments and Suggestions for Authors:

  • The manuscript analyzes the impact of oral health on daily activities in a population sample from Brazil. The justification of the study seems to be targeting the benefit that a NHS support could determine in primary dental care especially in deprived areas. The originality of the observation is not supported by a clear methodology. The planned enrolment seems to turn up in contingency plans and the overall data are gathered from stratified groups of the population with an unequal allocation by age. The primary and secondary outcomes are not clear and the discussion doesn't flag properly what the planned actions could be in the light of the observed results.

Response: The Unified Health System (SUS) revolutionized the health system in Brazil, whose guidelines began to oppose the current care model, based on a curative and medicalizing logic, and to propose family- and territory-centered care, consisting of disease prevention, promotion and health care actions, as described in the body of the text. However, this has not become the determining factor in the provision of oral health to the population. Thus, the justification for the study is not based exclusively on this approach.

To answer this question, the following was included in the text in lines 90 to 97: “The aim of this study was to develop a tool to improving of demand flow, by evaluating the impact of oral health on the daily activities of users of the Family Health Unit, by using the Oral Impacts of Daily Performances (OIDP) .

The demand flow in primary health care (PHC) is characterized by the user's attitude towards seeking health services to access and resolve their needs. The demand for health services can be spontaneous, when the user arrives at the health unit without having made an appointment; programmed, when health services are scheduled; and repressed, when people who need care are unable to access the health service.”

  • The paper shows a huge effort from an epidemiological point of view, but the planned actions in terms of public health promotions are not properly presented or explained.

Response: To answer this question, the following was included in the text in lines 279 to 288: “Combining the Coelho-Savasse Risk Scale with the OIDP Scale revealed important information about the impact on oral health and the possibility of scheduling patients for urgent care without affecting other dental services. There was a higher prevalence among females, which is also the group that most seeks oral health services. The proposed risk stratification for dental care can have significant practical implications for managing demand for oral health services, especially if high-risk patients can be seen more quickly. The introduction of the Wood classification to measure the urgency of need fordental care, based on risk and impact on oral health, offered a practical and potentially effective methodology for managing the demand for dental services.”

  • Comments on the Quality of English Language. Major changes

Response: The English of the entire text has been revised as requested.

*The National Health Strategy (NHS) was corrected in the revision of the English translation to Unified Health System (SUS).

Reviewer 3 Report

Comments and Suggestions for Authors

This study targets a very challenging subject, the impact of oral health on the daily activities.

The authors did a good job. The manuscript is well written and concise.

However, I have several comments to add:

Introduction:

The main objective of the study should be clearly written at the end of the introduction section.

Results section:

To be clearer, I suggest dividing this section into subparagraphs with subtitles.

Discussion section:

·      Please start your discussion section by presenting the main findings, which reply to the study aim.

·      Please highlight the strengths of the study, and discuss in more detail the weak points and limitations, particularly biases, and the generalizability of the results.

The conclusion is more a perspective than a conclusion. I would suggest improving that part.

As an additional task in the process of revising the manuscript, the text requires English language editing.

Comments on the Quality of English Language

As an additional task in the process of revising the manuscript, the text requires English language editing.

Author Response

Thank you for your valuable suggestions.

In the text, the requested revisions are in red and the English corrections are in blue

Comments and Suggestions for Authors:

This study targets a very challenging subject, the impact of oral health on the daily activities.

The authors did a good job. The manuscript is well written and concise.

However, I have several comments to add:

  • Introduction: The main objective of the study should be clearly written at the end of the introduction section.

Response: To answer this question, the following was included in the text in lines 90 to 97: “The aim of this study was to develop a tool to improving of demand flow, by evaluating the impact of oral health on the daily activities of users of the Family Health Unit, by using the Oral Impacts of Daily Performances (OIDP) .

The demand flow in Primary Health Care (PHC) is characterized by the user's attitude towards seeking health services to access and resolve their needs. The demand for health services can be spontaneous, when the user arrives at the health unit without having made an appointment; programmed, when health services are scheduled; and repressed, when people who need care are unable to access the health service.”

  • Results section: To be clearer, I suggest dividing this section into subparagraphs with subtitles.

Response: Thank you for your suggestion, we have divided the Results section into paragraphs with subheadings: “Data collection”; “Socioeconomic and clinical variables”; ‘Young people OIDP score”; “Adults OIDP questionnaire by family risk”; “The Wood classification”

  • Discussion section:
  • Please start your discussion section by presenting the main findings, which reply to the study aim.

Response: We have added some new information on the main results of the study. To answer this question, the following was included in the text in lines 283 to 292: “Combining the Coelho-Savasse Risk Scale with the OIDP Scale revealed important information about the impact on oral health and the possibility of scheduling patients for urgent care without affecting other dental services. There was a higher prevalence among females, which is also the group that most seeks oral health services. The proposed risk stratification for dental care can have significant practical implications for managing demand for oral health services, especially if high-risk patients can be seen more quickly. The introduction of the Wood classification to measure the urgency of need fordental care, based on risk and impact on oral health, offered a practical and potentially effective methodology for managing the demand for dental services.”

And lines 369 to 378:

  • Please highlight the strengths of the study, and discuss in more detail the weak points and limitations, particularly biases, and the generalizability of the results.

Response: To answer this question, the following was included in the text in lines 369 to 378: “The study contributed relevant information about the impact of oral health on a specific population and covered a wide age range, thereby providing valuable insights. The combination of the scales offered a thorough understanding of the impact on oral health, and observed a high prevalence in females, in agreement with previous research. The analysis of smoking identified a relevant risk factor, while the risk stratification proposal had the potential for optimizing dental care. The study had limitations such as the small sample of 430 participants, which made it difficult to generalize the results. The Covid-19 pandemic affected data collection hat migrated from home visits to on-demand collection, which may have impacted the representativeness of the sample, making it subject to bias.”

  • The conclusion is more a perspective than a conclusion. I would suggest improving that part.

Response: To answer this question, the following was included in the text in lines 381 to 392: “The conclusion is that creating a demand flow in the management of care at the Health Unit is an effective strategy for improving the tracking and monitoring of patients and highlighting the equity, universality, humanization and quality of the health service offered to the population. This has a positive impact on reducing demand on waiting lists and contributes to better planning of prevention, promotion and health care actions. The analysis showed that the adult population had a higher prevalence, confirming the importance of a self-perception tool that combines personal assessment with the individual's socio-economic assessment. Therefore, the implementation of a demand flow in health care management and the introduction of the Wood Scale represent significant advances in the improvement of the health services provided and in oral health actions, since they generate a higher number of consultations with better quality and equity of the services provided, promoting the well-being of the community served.

.”

  • Comments on the Quality of English Language. As an additional task in the process of revising the manuscript, the text requires English language editing.

Response: The English of the entire text has been revised as requested.

*The National Health Strategy (NHS) was corrected in the revision of the English translation to Unified Health System (SUS).

Round 2

Reviewer 2 Report

Comments and Suggestions for Authors

Fine with the Reviewing process. Editing  and  proofreading successfully completed. Eligible to get published.

Author Response

Thank you for your contribution.

In the text, the requested revisions are in red and the English corrections are in blue

Comments and Suggestions for Authors:

  1. Could you please clarify the technical terms used in the text to enhance its readability? For example, what do the terms 'risk 1', 'risk 2', and 'risk 3' refer to?

Response: An explanation of the terms "risk 1", "risk 2" and "risk 3" has been inserted in lines 135 to 139: “A sociodemographic questionnaire was used to collect patient identification data such as name, date of birth, gender, ethnicity, age, marital status, education, and family income; the ERF-CS was a useful instrument for classifying the vulnerability, and consequently, the risk factors called sentinels, stratified into 3 scores, and aggravating factors of the families registered. The score for each sentinel was proposed by the authors in accordance with the criteria of greatest impact on health and social life. Thus, 6 sentinels had score 3 (Bedridden; Physical Disability; Mental Disability; Low sanitation conditions; Severe malnutrition; Resident/room ratio greater than 1); 3 sentinels had score 2 (Drug addiction; Unemployment; Resident/room ratio equal to 1); 6 sentinels, score 1 (Illiteracy; Individual under six months of age; Individual over 70 years of age; Arterial hypertension; Systemic; Diabetes Mellitus; Resident/room ratio less than 1). The total sum would be classified as R1 (5 or 6); R2 (7 or 8); and R3 (greater than 9) [22].

  1. “could you explain what 'BHU' stands for in line 172?

Response: BHU has been corrected to FHU (Family Health Unit). (line 176).

The tool developed was based on a percentage of the population that was registered at an FHU.

  1. I had difficulty understanding the data presented in lines 270 to 277 in Table 4.

Response: The sentence was reformulated for better understanding (lines 274 to 276): “After the Wood classification of the study population (n=127), (W1=37, W2=55, W3=35), it was possible to prioritize the scheduling of patients with high urgency of need for dental treatment.”
